# Assessing Sustainability Priorities of U.S. Food Hub Managers: Results from a National Survey

**DOI:** 10.3390/foods12132458

**Published:** 2023-06-23

**Authors:** Haniyeh Shariatmadary, Sabine O’Hara, Rebecca Graham, Marian Stuiver

**Affiliations:** 1College of Agriculture, Urban Sustainability and Environmental Sciences (CAUSES), University of the District of Columbia (UDC), Washington, DC 20008, USA; sabine.ohara@udc.edu; 2Institutional Assessment, University of the District of Columbia (UDC), Washington, DC 20008, USA; rebecca.graham@udc.edu; 3Green Cities Programme, Wageningen University and Research (WUR), 6708 PB Wageningen, The Netherlands; marian.stuiver@wur.nl

**Keywords:** food hubs, environmental sustainability, social equity, food security, food hub managers’ survey, local food systems, sustainable agriculture

## Abstract

Food hubs have emerged as innovative alternatives to the conventional United States food system. As aggregators of small local farms, food hubs hold the potential to transform food production, distribution, and consumption, while fostering environmental sustainability and social equity. However, assessing their contributions to environmental sustainability and social equity is challenging due to the diverse structures and practices of U.S. food hubs. This study presents the findings of a national survey of food hub managers conducted in 2022 to assess the sustainability objectives and practices of food hubs across the United States. Our survey questions were designed based on a comprehensive framework of social and environmental sustainability criteria. Our results reveal that food hubs make valuable contributions in supporting small producers and providing healthy local food options. However, there is room for improvement in their environmental sustainability practices, as they only meet 47% of the defined environmental sustainability goals. Addressing food insecurity is a high priority for food hubs, although not their top priority, and many offer fresh food access to low-income households. Food hubs also contribute to environmental sustainability by reducing food transportation, promoting healthy food production methods, and minimizing waste. While food hubs meet 67% of the defined social sustainability goals, there are opportunities for improvement in reaching important institutional stakeholders and enhancing consumer education on healthy nutrition and lifestyles. Expanding technical assistance for farmers is also critical. By addressing these opportunities for improvement, food hubs can drive progress towards a more resilient and equitable food system in the United States.

## 1. Introduction

The conventional food system is confronting a multi-dimensional crisis, stemming from various environmental and social issues, such as nutrient loss [1,2,3], climate change [4,5,6,7], biodiversity loss [8,9,10,11], reduced biophysical health [12,13], soil depletion [14], poor animal welfare standards [15,16], food access disparities, and health concerns. The overproduction of food exacerbates these negative externalities, leading to a staggering loss of one-third of the food produced. In recent years, food hubs have emerged as a promising alternative to the conventional U.S. food system, which is characterized by large-scale, centralized food production in rural areas, and long vulnerable food supply chains to provide food access to consumers in urban and metropolitan areas [17,18,19,20].

Food hubs, as defined by the USDA, serve as aggregators of small producers, offering an organized business management structure to streamline the aggregation, storage, processing, distribution, and marketing of locally and regionally produced food [21]. These innovative alternatives aim to enhance the efficiency, resilience, sustainability, and accessibility of the U.S. food system [22]. As a result, food hubs appear to be well positioned to improve the sustainability of the current food system [17,23,24,25] and have the potential to drive transformative change by making multifaceted contributions to the social and environmental sustainability of the U.S. food system. However, despite their perceived potential, there is a need for empirical evidence to support the proposition regarding the social and environmental benefits of food hubs. Our study aims to fill that gap by assessing the impact of food hubs on specific social and environmental sustainability indicators, which help determine their potential to effect transformative change.

The diverse structures and practices of food hubs pose challenges in evaluating their social and environmental sustainability contributions [26]. To our knowledge, our earlier work remains the only study to date that has examined the sustainability goals of U.S. food hubs in a systematic manner [27]. While this study, which relied on publicly available information, offered valuable insights, it also resulted in some ambiguous findings. For example, limited published information on food hubs’ sustainability goals does not necessarily imply disinterest in pursuing these goals. Notably, the previous study revealed that food hubs seemed to pay little attention to food insecurity or promoting water-saving production methods. To address the potential ambiguities of our previous findings, this paper reports on a follow-up study which used a survey approach to directly investigate the self-reported sustainability objectives of food hub managers.

Building on the previous literature on food system sustainability, we designed our survey questions based on six broad social and environmental sustainability indicators. We used two social sustainability indicators (1) increasing access to fresh and healthy food, and (2) reducing food insecurity. We also use four environmental sustainability indicators: (1) promoting healthy food production, (2) reducing food transportation, (3) minimizing food waste and loss, and (4) improving efficient water management. Each of these broad questions include sub-questions to further assess contributions made by food hubs in our sample to these six categories. When designing our survey, we ensure alignment with our previous study to enable comparison of the results.

The survey methodology offers several advantages in assessing sustainability objectives of food hubs [28]. First, a survey facilitates the direct collection of data from food hub managers, providing more in-depth insights into their priorities. For instance, our survey explicitly inquired about the sustainability goals and priorities of the food hub managers, while our previous study relied on indirect information from mission statements and related materials. Second, survey questions can be tailored to collect more specific information from individual food hubs, such as evaluating progress toward their sustainability objectives. For instance, the survey allowed us to obtain better information about a food hub’s contribution to reducing food transportation by asking about the number of collaborating farms and their proximity to the food hub. This level of detail is not consistently available in publicly accessible data for the food hubs in our sample. Lastly, by conducting a survey, we were able to gather up-to-date and consistent information across our entire sample of food hubs, whereas some of the publicly available data was somewhat outdated and may have referred to past practices that no longer accurately reflected current practices.

To quantify the results of the survey responses and aggregate them across all food hubs for our six sustainability indicators, we calculated an achievement score for each of our six indicators (two social and four environmental ones). The achievement score, ranging from 0% to 100%, measures the degree to which the food hubs in our sample have collectively accomplished our defined social and environmental sustainability objectives. We then use the underlying achievement scores to evaluate the aggregated scores for the sustainability contributions of all of the food hubs in our sample.

Apart from our study, the National Food Hub Survey [26] appears to be the only other national-level survey of food hubs in the United States identified in the literature. The most recent survey results of the USDA National Food Hubs survey date back to 2019. This survey was first launched in 2013 and is conducted biennially. Its primary aim is to offer insights into the operations, trends, and challenges U.S. food hubs face. The survey is not expressly designed to focus on the sustainability contributions of food hubs. Instead, it focuses more on the financials, business performance, supply chains, and support services available to the local producers within a food hub. While the survey addresses some of the contributions of food hubs to increased access to healthy, locally-produced food, its focus is not expressly on the social and environmental sustainability contributions of food hubs. Our study thus fills a critical information gap by providing information about the sustainability contributions of food hubs, thereby offering insights into the potential sustainability contributions of a more localized and decentralized food systems model.

The strength of our study lies in its comprehensive and consistent framework, which encompasses both social and environmental sustainability criteria, broadly defined. This approach enables us to create a quantitative ranking of the sustainability contributions of the food hubs in our sample based on their contributions to our defined sustainability objectives.

This paper will first outline our methodology for surveying food hub managers, detail our sample selection, discuss our survey design, and analyze our survey results and the techniques employed for our data analysis. We then discuss our findings regarding the sustainability objectives and practices of the food hubs in our sample and highlight commonalities and differences between the survey results and previous studies. Finally, we place our findings in the broader context of the literature, highlighting areas where food hubs have successfully achieved sustainability goals and identifying opportunities for further advancements in creating a more resilient and equitable food system. We conclude with a brief discussion of future research needs and the wider significance of our findings for a more sustainable food system in the United States.

## 2. Methodology

To assess the social and environmental sustainability contributions of Food Hubs across the U.S., we conducted a survey of food hub managers. Our sample included Food Hubs with various organizational characteristics, including private sector businesses, non-profit organizations, and cooperatives, and came from various regions across the United States. The target population for our survey consisted of the managers of all food hubs operating within the United States.

### 2.1. Sample Selection

The basis for our survey sample was the USDA Food Hubs Directory, which consists of a list of over 230 Food Hubs. In determining the actual size of our survey population, we found that data about the actual number of food hubs operating in the United States differs from the reported sample size of the USDA Food Hubs Directory. While the directory lists 230 food hubs, a review of available contact information and online information about the listed food hubs revealed that several of them were no longer in operation. Additionally, as food hubs self-identify and self-report their information for inclusion in the directory, we found that some did not align with the definition of a food hub. For example, some operated solely as food processors and not as aggregators for small food producers. We excluded all non-qualifying and inactive food hubs from our survey sample and concluded that the basis for our survey consisted of 150 active food hubs.

Our survey tool was structured around our previously defined sustainability indicators, consisting of two social sustainability indicators and four environmental sustainability indicators. The survey design was reviewed and approved by the Institutional Review Board of the University of the District of Columbia. After an initial effort to obtain contact information for all active food hubs, we distributed a Qualtrics-designed survey [29] to 150 food hub managers. The survey remained open from May 2022 to the end of August 2022. In addition to soliciting participation via email, we also contacted food hub managers by phone. Some agreed to complete the survey over the phone, with the conversations providing valuable supplementary information related to our survey questions. We received a total of 34 surveys, whereby 30 of them were fully completed without any gaps, and 4 were partially completed. This corresponds to a response rate of 20%. Several other food hub managers cited time constraints and staffing shortages as reasons for their non-participation in the survey.

### 2.2. Survey Design

The questions included in our survey regarding the social and environmental sustainability goals of U.S. Food Hubs sought to capture key elements of a sustainable food system as outlined in the literature. Our questions were based on the underlying premise that a sustainable food system is “a food system that delivers food security and nutrition for all in such a way that the economic, social and environmental bases to generate food security and nutrition for future generations are not compromised” [30,31]. All food hub managers were asked to respond to the same set of questions.

Our assessment of the social sustainability goals of the Food Hubs in our sample was based on two aspects of the above cited definition, namely providing sustainable access to nutritious food for both regular consumers and underserved consumers dealing with food insecurity. Our specific questions were (1) Are you committed to increasing access to fresh and healthy food for regular consumers? (2) Are you committed to addressing food insecurity for underserved consumers? The latter takes into account the health and productivity implications of food insecurity [32], which could lead to significant social disparities [33].

Our study of the environmental sustainability contributions of Food Hubs focused on the impact of food production systems on the environmental resources, comprised air, water, and soil resources [34]. Long-distance transportation, cold chains, and storage associated with distant food consumer and producer locations contribute to greenhouse gas emissions and energy use [31,35,36,37]. Moreover, adopting good agricultural practices that limit the use of pesticides and care for soil quality and water use also impacts environmental resources [38,39,40]. Food waste at various stages of food production, distribution, and consumption is another significant factor that impacts the environment [31,41] through greenhouse gas emissions, water and energy consumption, and occupying landfill space [42,43,44]. Water use in crop production represents the bulk of freshwater use, and improved agricultural practices have considerable potential for improving fresh water savings and improved water management [45]. Based on these observations, we designed our environmental sustainability questions around four indicators: (1) promoting healthy food production, (2) reducing food transportation, (3) minimizing food waste and loss, and (4) improving efficient water management.

Our sustainability indicators are well aligned with the Sustainable Development Goals (SDG) of the United Nations [46]. Table 1 shows the link between our sustainability indicators and UN’s SDGs. SDG 11 and SDG 13 are broadly aligned with social and environmental indicators. The underlying sub-indicators (on the left column) are specifically linked to relevant SDGs.

Our survey questions also addressed basic background information about the food hubs, including: (a) their organizational structure: survey participants were asked to identify their food hub as a business, non-profit organization, or cooperative; (b) their scope of operation: survey participants were asked about the number of farms with which their food hub collaborates, the types of fruits and vegetables offered, the revenue generated in 2020, and the number of paid staff members and volunteers; (c) their operational model: survey participants were asked to specify the operational model of their food hub by indicating whether they engage in direct sales to customers, wholesale operations, or a hybrid model; and (d) their location: survey participants were asked about the location of their food hub, identifying whether it is situated in an urban, suburban, or rural area.

To elicit responses regarding the degree to which a Food Hub contributes to our six sustainability indicators, the survey asked food hub managers about their objectives relative to each indicator, using a four-point scale to rate the importance of each: 0 indicated no contribution; 1 a weak contribution; 2 some contribution; and 2 a strong contribution. Additionally, the survey posed specific questions related to each broad sustainability indicator to assess the significance of each indicator and gain a better understanding of how food hub managers strive to achieve the defined objectives in practice. Appendix A Table A1 presents the complete set of survey questions, along with their response options and corresponding ratings for each sustainability indicator.

### 2.3. Data Analysis

The survey data collected included frequency distributions for a scale from 0 to 3, with 0 indicating “not important” and 3 indicating “very important.” A higher concentration of responses for larger values (such as 2 and 3) implies a greater contribution toward a particular indicator by the food hubs in our sample. To show a single number representing the average contribution of all food hubs to each individual sustainability indicator, we calculated an average achievement score according as follows:(1)$$Average\;Achievment\;Rate=\frac{1}{3}\omega_{1}+\frac{2}{3}\omega_{2}+\omega_{3}.$$

In Equation (1), $\omega_{1},\;\omega_{2},$ and $\omega_{3}$ are the frequency shares of responses 1, 2, and 3, respectively. Assuming that 30 food hubs responded, and 5 food hubs choose 0, 10 choose 1, 10 choose 2, and 5 choose 3, the achievement rate would be calculated as follows:

$\omega_{1}=\frac{10}{30},\;\omega_{2}=\frac{10}{30},$ and $\omega_{3}=\frac{5}{30}$. This implies an average achievement rate of $\frac{1}{3}\times \frac{10}{30}+\frac{2}{3}\times \frac{10}{30}+\frac{5}{30}=50\%$, meaning that the food hubs collectively meet 50% of the maximum contribution toward the particular sustainability goal. The average achievement rate is by structure between 0%, when $\omega_{1}=\omega_{2}=\omega_{3}=0$, and 100% when $\omega_{1}=\omega_{2}=0$ and $\omega_{3}=1$. A value of 100% indicates that all food hubs achieve a perfect score of three (3), while a score of zero (0) indicates that none of the food hubs have made any contribution toward achieving a particular sustainability indicator. A higher value, therefore, reflects a greater contribution to achieving a particular sustainability indicator.

## 3. Discussion of Results

The survey responses received from 30 food hub managers provide a comprehensive representation of the diverse organizational structures, operational models, and geographic locations of food hubs across the United States. Figure 1 shows that the surveyed food hubs (located in states highlighted in red) are broadly distributed throughout the country and capture a wide range of climate zones. The somewhat higher representation of the population centers along the U.S. east coast is to be expected. The food hubs in our sample, from the population of all active food hubs we attempted to reach out to, participated voluntarily. Notably, the surveyed food hubs are located in states that account for approximately three-fourths of the total food hubs in the population (shaded in red areas in Figure 1).

### 3.1. Descriptive Statistics

Table 2 presents the descriptive statistics for our sample of 30 food hubs included in our survey. About 65% of the food hubs in our survey are businesses, 25% are non-profit organizations, and 10% are cooperatives. About 30% of the surveyed food hubs work with 20 or fewer farms, 40% collaborate with 21 to 50 farms, and 30% partner with more than 50 farms. About 40% of the food hubs generate USD 500,000 or less in annual revenue, about 30% generate between USD 500,000 and USD 1 million, and about 30% generate more than USD 1 million in revenue. The prevailing revenue model among the surveyed food hubs is direct sales to end consumers, with 40% of the food hubs reporting that their primary focus is direct sales to households through various distribution models. Another 47% report having a hybrid model of direct sales to consumers and sales to wholesalers. Only 13% sell primarily to wholesalers. All of the surveyed food hubs report a mix of full-time and part-time staff as well as volunteers. In general, the paid workforce of food hubs is small, and several rely on technology to address the significant coordination tasks associated with the steady and reliable operation of a food hub. Our data suggest that reliance on volunteers is not a significant factor. Instead, some food hubs prioritize efficiency and a small, well-trained multitasking workforce, while others rely on part-time and seasonal labor to support their operations.

As illustrated in Table 2, the geographic distribution of the food hubs in our survey sample is diverse and primarily includes food hubs in urban and metro areas, with a small number located in rural areas. The descriptive statistics of the surveyed food hubs in Table 2 support the randomness of the responding sample and the representativeness of the sample for the target population. This together with the reasonably high response rate strengthen the reliability of the findings based on our survey results, which we present next.

### 3.2. Sustainability Performance Assessment

The distributional results of our survey of food hub managers are summarized in Appendix A Table A2. The frequencies are reported as percentages, which reflect the proportion of scores of a particular value. For example, 83% corresponds to 25 responses out of 30. The rankings of each survey question and the achievement scores (as explained previously) for each sustainability criterion are listed in the right-hand column of the table. Using distributional data from Table A2, Table 3 calculates the confidence interval for each indicator. The table also reports the underlying components used for the calculation of the confidence interval, including the expected mean, standard error, and the Z-score. The generally low able 3standard errors for most of the indicators that result in large Z-score values shows the reliability of the results for assessing our research questions.

**Increasing Access to Fresh and Healthy Food.** Our analysis indicates a significant commitment by food hubs to increase access to fresh and healthy food for local consumers. The confidence intervals associated with the set of indicators under increasing access to fresh and healthy food are aligned with the hypothesis that food hubs make strong contribution to this social sustainability indicator (the first two indicators). Likewise, the associated achievement scores indicate that over 90% of the surveyed food hub managers place significant importance on their contribution to improved food access for their local customers. This underscores the role food hubs can play in enhancing the social and health benefits associated with increased access to healthy food. These benefits have been widely recognized in the literature [32,33,47]. However, we also found that food hubs’ performance was weaker in providing fresh and nutritious food to schools and colleges, suggesting opportunities to further improve the social sustainability contributions of food hubs with respect to local organizations and institutions as compared to individual customers. The findings are confirmed in the literature, which also suggests that students have more limited access to healthy and fresh food compared to other consumer groups, which has critical long-term implications [48,49].

In addition, our study demonstrates that when it comes to consumer education, the food hubs’ contributions are weak. This shows that food hubs have the potential to play a more proactive role in providing training programs for their customers. Our findings suggest that 20% of the food hubs in our sample consistently offer cooking classes, healthy eating programs, and education on reducing food waste. These findings also align with the existing literature that emphasizes the importance of early and frequent efforts to provide nutrition education, food tastings, and other strategies to change attitudes, and behaviors toward more healthy and active lifestyles, which are essential [50,51] especially among children and young adults [52,53]. Food hubs could play an important role by augmenting their contributions to social sustainability though expanded educational initiatives for younger populations.

**Addressing Food Insecurity.** The confidence intervals for food insecurity indicators show that food hubs make some contribution and possibly a strong contribution. Our results indicate that food hubs contribute significantly to addressing food insecurity within the communities where they are located. The food security objective in our survey achieved a score of 76%, which is significantly higher than the 35% score reported in our previous study based on publicly available information. The difference may suggest that while food insecurity is a high priority for food hub managers, they may not actively vocalize their efforts in this regard. It is noteworthy that non-profit food hubs that might be expected to expressly focus on alleviating food insecurity, as suggested in the literature [54,55], constituted only a quarter of our sample. Most of the food hubs in our survey sample were businesses indicating that alleviating food insecurity is a high priority for food hubs regardless of their organizational model.

Furthermore, we found that the majority of food hubs surveyed intentionally selected a location that is easily accessible to low-income households. This supports previous findings in the literature [56,57,58] and suggests that our previous analysis may not have sufficiently picked up on the nuances of the relative distance to low-income households. The accessibility indicator received a higher score of 70% which is significantly higher than the 53% score in our previous study.

Additionally, our survey data allowed us to explore the diverse strategies food hubs utilize to facilitate access to food for underserved populations within their communities. Two-thirds of food hubs reported donating unsold produce [59,60,61]; one-third accept SNAP, WIC, or other food assistance (see [62] for a critical discussion); others provide subsidized fruits and vegetables, or offer nutrition education programs aimed at preparing healthy meals on a budget. These findings indicate that food hubs adopt a range of strategies to address food insecurity in their communities, while actively collaborating with other organizations focused on improving food security, promoting healthy lifestyles, and reducing food waste.

**Promoting Healthy Food Production.** Food hubs can play a crucial role in fostering sustainable food systems by advocating for more sustainable and healthy food production practices. Our study indicates that 50% of the food hubs surveyed actively promote such practices and seek to contribute to both social and environmental sustainability through reduced pesticide- and herbicide-use and related practices. There is, however, considerable potential for improvement. Only 13% of the hubs in our sample offer capacity-building opportunities to their member farms and actively promote sustainable soil management, nutrient management, waste reduction, and pest management practices. Prior research also suggests that urban farmers especially expressed a significant interest in receiving more technical assistance on various farming aspects [63]. This highlights the importance of these services and the opportunities food hubs have in providing them. These findings suggest that food hubs can enhance their role in actively promoting and providing technical assistance to their member farms on sustainable farming practices. Expanding these services could significantly enhance the environmental sustainability contributions of food hubs [64].

**Reducing Food Transportation.** Our hypothesis tests substantiate a mildly significant contribution of food hubs to environmental sustainability through reducing food transportation. This is achieved by aggregating fruits and vegetables from multiple local farms, thereby minimizing the environmental impact associated with long-distance transportation. In particular, the survey results provide compelling evidence. Among those surveyed, an impressive majority of over 70% reported working with at least 20 farms. Notably, two-thirds of these farms are located within a 100-mile radius of the food hubs. Additionally, we found that more than half of the surveyed food hubs actively offer transportation services to their members, further reducing the overall food transportation needs of their local community customers. These findings suggest that food hubs can play an important role in promoting a more sustainable food system that reduces the carbon footprint associated with food transportation [37,65].

**Reducing Food Waste and Loss.** Our survey results confirm that food hubs play a significant role in contributing to reduced food waste and loss, chiefly by reducing the time and distance between farm and fork. The commitment of the food hubs to source from local farmers reduces the need for cold chain and transportation-related losses [65,66]. Consistent with our previous findings, food hubs play an active role in reducing food waste, not only as an environmental sustainability contribution [41], but also as a social sustainability contribution. By donating unsold food, the food hubs also contribute to reducing food insecurity. Moreover, we found that more than 50% of the food hubs surveyed compost their food waste [67,68,69]. This is far higher than the 12% we identified in our previous study. This indicates that the food hubs we surveyed are taking active steps towards reducing the negative environmental and social impacts associated with food waste and loss. However, there is still room for improvement and only a few of the surveyed food hubs offer education and training programs focused on food waste reduction and composting.

**Promoting Water Management.** Our survey results suggest that the surveyed food hubs are making an effort, albeit statistically weak, to promote better water management. In our previous study, this was the weakest of the sustainability indicators. While our previous study revealed almost no contributions to improved water management, our survey revealed that about 30% of the food hubs in our sample actively work with their member farms to improve water management and utilize more water saving food production techniques, including drip irrigation, hydroponics, and aquaponics. These soilless production methods can offer substantial water savings and advance the more efficient use of a potentially scarce resource. Other studies also indicate that farmers, especially in some European countries, expressed a strong preference for learning more efficient water management techniques, including receiving training and certification in water management [70].

While our survey focused on the social and environmental sustainability contributions of U.S. food hubs, food safety practices too can have significant implications for the efficiency, reliability, and sustainability of food systems. Food safety practices are not simply influenced by scientifically determined standards, but also by culturally determined ones. This also implies that food safety standards may accrue different benefits to different groups of food producers and consumers. These differences must be taken into account when food safety standards are considered to safeguard food hubs and food supply chains against foodborne illnesses [71] and other food safety risks [72,73]. Future research might explore common ground between diversified structures of food hubs and the diversified needs for culturally relevant food safety practices.

### 3.3. Aggreagted Results

To quantify the contributions of food hubs based on our broad-based indicators, we calculated the sum of the impacts of the food hubs (Figure 2). The full responses obtained from 30 food hub managers in our survey represent a diverse range of organizational structures, operational models, and geographic locations of food hubs across the U.S. Figure 1 shows that the surveyed food hubs are distributed across the U.S. and capture a wide range of geographic locations and climate zones. Our cumulative results indicate an overall achievement level of sustainability goals of 57%; there is a greater emphasis on social sustainability indicators of 67%, compared to a cumulative achievement score of 47% overall for the environmental sustainability indicators. This means that food hubs in our sample indicate that they achieve 67% of social sustainability indicators and 47% of environmental sustainability indicators. The lower score for the environmental sustainability indicator is associated with a relatively weaker emphasis on promoting sustainable production practices compared to the social sustainability commitments of the food hubs.

Our survey confirms the substantial contributions food hubs make to improving access to fresh, locally-sourced food. The social sustainability commitment of the food hub managers in our sample is particularly evident in their efforts to improve food access and reduce food insecurity. About 75% of the surveyed food hubs are intentionally located in low-income neighborhoods in an effort to improve food access and food security among households most in need. Food hub managers also indicated their commitment to source from local farms, thus contributing to both environmental and social sustainability. By forming strong bonds with their local farms and actively addressing food insecurity within their communities, food hubs contribute significantly on both scores.

### 3.4. Limitations of the Study

Our study represents a pioneering effort in examining the social and environmental sustainability contributions of U.S. food hubs through a survey-based approach. However, to strengthen the validity and generalizability of our results, further research is warranted. As discussed in the methodology section, our survey sample consisted of 30 respondents, constituting approximately 20% of the active food hubs in the US Food Hubs survey. Although our sample encompassed diverse and geographically dispersed food hubs, it is important to acknowledge that the actual number of active food hubs may be considerably larger. Thus, a larger and more representative sample would provide a more comprehensive understanding of the subject.

While our efforts in including a diverse range of respondents help mitigate concerns regarding bias often found in small homogenous samples, it is crucial to conduct additional surveys to enhance the reliability of our findings. Moreover, a longitudinal assessment of the sustainability contributions of food hubs over time would be invaluable in strengthening the representativeness of our sample and facilitating further statistical analysis. By addressing these limitations, future research endeavors will be better equipped to substantiate and expand upon the findings of this study, allowing for a more comprehensive understanding of the social and environmental impact of U.S. food hubs.

## 4. Conclusions

The aim of this study was to assess the sustainability priorities of food hub managers across the United States based on their contributions to six overarching social and environmental sustainability indicators. Given persistent questions about the sustainability contributions of food hubs, both from the perspectives of producers and consumers, we conducted a survey of 30 food hub managers, representing approximately 20% of all active food hubs listed in the USDA Food Hubs directory. Based on the rigorous assessment of our survey results, we conclude that US food hubs make considerable sustainability contributions measured in terms of two social and four environmental sustainability indicators. The insights gained from our study provide not only valuable information about the current sustainability contributions of U.S food hubs, but also about intervention strategies that promise further improvements. The environmental sustainability contributions of food hubs particularly offer promising opportunities through changes in the capacity-building programs the food hubs offer. These improvements promise to be both achievable and impactful.

## Figures and Tables

**Figure 1 foods-12-02458-f001:**
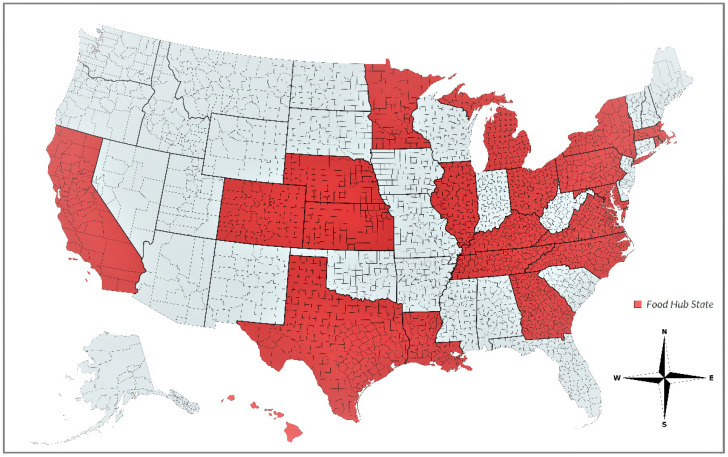
Geographical Distribution of Sampled Food Hubs: States with Participating Food Hubs in the Survey Shaded in Red. Created with www.mapchart.net (accessed on 26 March 2023).

**Figure 2 foods-12-02458-f002:**
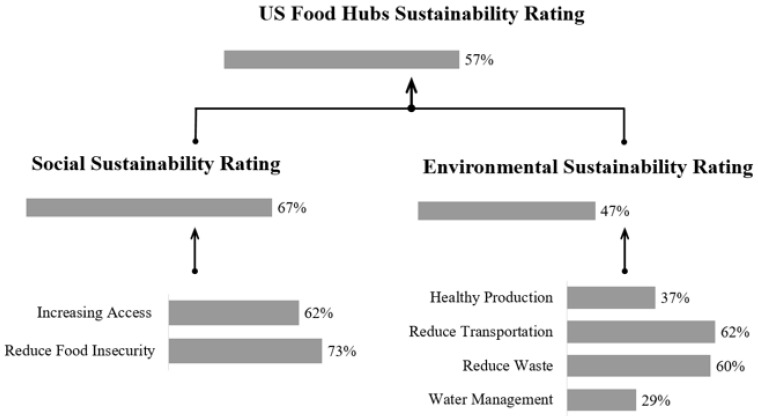
Aggregate Sustainability Rating of Selected U.S. Food Hubs based on the social and environmental sustainability framework.

**Table 1 foods-12-02458-t001:** Mapping Sustainability Indicators Used in the Paper (left column) to the UN’s Sustainable Development Goals (right column).

Sustainability Indicators	Sustainable Development Goals (SDGs)
Social	SDG 11 ‘sustainable cities and communities’
(1) Increasing access to fresh and healthy food	SDG 3 ‘good health and well-being’ SDG 12 ‘responsible consumption and production’
(2) Addressing food insecurity	SDG 2 ‘zero hunger’ SDG 3 ‘good health and well-being’ SDG 10 ‘reduced inequality’ SDG 1 ‘no poverty’
Environmental	SDG 13 ‘climate action’
(1) Promoting healthy food production	SDG 12 ‘responsible consumption and production’ SDG 16 ‘clean water and sanitation’ SDG 15 ‘life on land’ SDG 14 ‘life below water’
(2) Reducing food transportation	SDG 7 ‘affordable and clean energy’
(3) Minimizing food waste and loss	SDG 16 ‘clean water and sanitation’ SDG 15 ‘life on land’
(4) Improving efficient water management	SDG 16 ‘clean water and sanitation’

**Table 2 foods-12-02458-t002:** Descriptive Statistics of Food Hub Characteristics in the Sample (*n* = 30).

Organizational Structure		Farm Membership Size				Annual Revenue in 2020	
A business	19	Single farm	1			less than USD 50,000	1
A cooperative	3	2 to 10 farms	1			USD 50,001 to 100,000	3
A non-profit organization	8	11 to 20 farms	7			USD 100,001 to 500,000	10
		21 to 50 farms	12			USD 500,001 to 1 million	8
		more than 50 farms	9			More than USD 1 million	8
Operational Model		Employment Composition	full-time	part-time	volunteers	Location	
Direct sales to customers	12	less than 5	13	12	2	In a rural area	5
Hybrid	14	6 to 10	11	5	3	In a somewhat urban area	7
Wholesale	4	11 to 20	4	1		In a suburban area	6
		more than 20	1	2	1	In a very urban area	12

**Table 3 foods-12-02458-t003:** Hypothesis Tests of the Survey Results of Food Hubs’ Contributions to Social and Environmental Sustainability.

Expected Value	Standard Error	Z-Score	Confidence Interval	
**Increasing Access to Fresh and Healthy Food**				
How important is increasing access to healthy and fresh food for local consumers for your food hub?				
2.80	0.09	32.21	2.63	2.97
How do you rate the importance of local consumers for your food hub’s operation?				
2.77	0.14	19.92	2.49	3.04
How do you rate the importance of schools and colleges for your food hub’s operation?				
1.67	0.21	8.04	1.26	2.07
Does your Food Hub offer convenient and effective means for the direct sale of products to consumers?				
1.70	0.27	6.26	1.17	2.23
Do you offer any training programs for your consumers?				
1.27	0.23	5.49	0.81	1.72
**Addressing Food Insecurity**				
How important is reducing food insecurity for your food hub?				
2.27	0.16	13.92	1.95	2.59
In your opinion, is your food hub located in an area that is accessible to a low-income households?				
2.10	0.17	12.19	1.76	2.44
**Promoting Healthy Food Production**				
Does your food hub require any of the following from supplying farmers: organic production, chemical-free production, or good agricultural practices (GAPs)?				
1.60	0.23	6.99	1.15	2.05
Does your food hub offer trainings for growers?				
0.63	0.20	3.21	0.25	1.02
**Reducing Food Transportation**				
How many farms do you work with?				
2.20	0.12	18.45	1.97	2.43
Where are farms that supply the majority of your produce located relative to your food hub?				
1.97	0.17	11.36	1.63	2.31
Do you offer food transport or delivery services to the door steps of households?				
1.60	0.27	5.86	1.06	2.14
How important is selling to local food businesses for your food hub?				
1.63	0.22	7.31	1.20	2.07
**Minimizing Food Waste and Loss**				
Does your food hub actively pursue donating unsold food?				
2.10	0.25	8.37	1.61	2.59
Does the Food Hub actively compost waste produce?				
1.50	0.27	5.48	0.96	2.04
**Promoting Water Management**				
Do some of your farmers grow fruits and vegetables using hydroponic or aquaponic methods?				
0.87	0.11	7.68	0.65	1.09

Note: The last two columns represent the lower and upper bounds of the confidence interval at a 95% significance level. The hypothesis test implies if the confidence interval contains 2.5 and greater, food hubs make a strong contribution, between 1.5 and 2.5, some contribution, and between 0.5 and 1.5 weak contribution and less than 0.5, no contribution.

## Data Availability

The data used to support the findings of this study can be made available by the corresponding author upon request.

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
