# Peer review of "Assessing Sustainability Priorities of U.S. Food Hub Managers: Results from a National Survey"

_foods, 2023, doi:10.3390/foods12132458_

Round 1

Reviewer 1 Report

This article “Assessing Sustainability Priorities of U.S. Food Hub Managers: Results from a National Survey” presents the findings of a national survey of food hub managers conducted in 2022 to assess the sustainability objectives and practices of food hubs across the United States, and the survey questions were designed based on a comprehensive framework of social and environmental sustainability criteria. The paper presents high research significance, and the contents meet the reader's interest of Foods. There are some shortcomings that need to be further explained or improved.

Comments:

Q1. Figure 1, Why choose the food hubs in these states? Did other states also have food hubs?

Q2. What do the authors consider might account for the differences between this survey and previous studies in Table 4.

Q3. In Table 4, we can find that the evaluation indicators of this paper are roughly the same as the previous contents, then how to reflect the significance of the innovation of this article.

Q4. The expressions of the conclusion are too tedious, so it is suggested to put part of it in the discussion of results, and only one paragraph with summary language can be reserved for the conclusion.

Author Response

Dear Reviewer,

Thank you so much for your supportive feedback and posing these constructive questions. Please find below our responses addressing each of your inquiries.

Regarding your first question, we have provided additional clarification in subsection 2.1 and the introduction of section 3 to address concerns about the selection of food hubs in our sample. It is essential to note that our study did not involve the deliberate "choosing" of a subset of food hubs for inclusion. Instead, we proactively reached out to all active food hubs via phone and email, inviting them to participate. The paper presents the results obtained from the final number of respondents, which constitutes more than one-fifth of the entire population of food hubs in the United States. To provide a more comprehensive perspective, we have included information on the states where the food hubs in our sample were located. These states represent three-fourths of the total population of food hubs. Furthermore, we acknowledge that 25% of the food hubs that did not respond were located in states without any participants in our sample.

Regarding your second question, we address the differences between our findings in this paper and previous studies in the discussion of results. However, we have decided to remove Table 4 from this draft to avoid potential distractions and ensure a clear focus on the main survey results.

Regarding your third question, we recognize that this paper's approach, using a direct survey of food hub managers, provides a more effective means of gathering relevant data for assessing sustainability priorities compared to previous studies that relied on publicly available information. The alignment between findings based on publicly available information and our survey results indicates the reliability of the former, given their larger sample size. However, the disparities between the two sources of information underscore the limitations of relying solely on publicly available data, as they may not fully capture the perspectives of food hub managers. Removing the table in question from the draft helps maintain clarity and focus.

Finally, we have shortened the conclusion and shifted some of the more detailed discussions to the previous section to ensure a concise and impactful closing while still providing relevant insights and directions for future research.

Thank you for raising these questions, and we appreciate the opportunity to address them and make the necessary revisions in response.

Sincerely,

Haniyeh Shariatmadary,

on behalf of the team of authors.  

Reviewer 2 Report

The paper is interesting, very well written and nicely laid out. It.  The authors studied the findings of a survey responded by merely 34 active hubs. I guess that getting these responses should have been hard but nonetheless, the validity of the study is questionable.

The questions can be also objected. One can guess that providing affordable foods is more important for a food hub than promoting the good use of water. Therefore the results are very predictable.

This paper can be more suitable for another type of publication such a report for governmental agencies or for an association of food hubs. 

I really enjoyed reading the paper and I see some merit in it, but the authors must focus on insights that the surveys can provide. 

Author Response

Dear Reviewer,

Thank you for your valuable comments and supportive feedback. We have carefully considered and incorporated your suggestions into the new draft of the manuscript.

One notable improvement is the enhancement of our analysis through the inclusion of hypothesis testing for each sustainability indicator. This has resulted in small standard errors, high Z-scores, and narrow confidence intervals, which confirm the reliability of our analysis. The relevant statistical information has been incorporated into Table 3, while the previous table has been moved to the appendix (Table A2). Additionally, we have provided details on the response rate in section 2.1 and emphasized the randomness of our sampling process, supported by the inclusion of diverse food hubs and coverage of multiple states in our sample, accounting for 75% of all food hubs.

In the paper, we have emphasized the significance of the sustainability aspect in food hubs, which extends beyond the affordability of providing food to consumers. While economic aspects have been the focus of previous literature, our study places equal, if not greater, importance on the social and environmental sustainability of food systems and food hubs. We have explicitly established the connection between our framework and the United Nations' Sustainable Development Goals, highlighting the relevance and significance of this perspective.

Regarding the target audience, we acknowledge that our paper is not limited to researchers but also appeals to policy makers and practitioners involved in shaping local food systems. The paper's focus on a key topic within local food systems, coupled with the utilization of an established methodology, makes it highly suitable for publication in the Foods journal. By adhering to rigorous research methods and addressing a significant aspect of food systems, our paper aligns well with the journal's scope and contributes valuable insights to the broader academic and professional community.

Finally, we have sharpened the explanation of the survey results and emphasized the main findings in the conclusion, thereby contributing to the emerging literature in this field.

We sincerely appreciate your comments and supportive feedback, as they have greatly contributed to the improvement and refinement of our manuscript.

Sincerely,

Haniyeh Shariatmadary

on behalf of the team of authors.  

Reviewer 3 Report

Article written by authors is the potential to publish in this journal but after revision.

English language is very verbose, it should be correct.

Authors did't try to start from base, they directly jump in every part of the article.

Spacing problem occurs within the text.

Attached PDF contains some very important issue, these should be solve before sending back to the journal. 

It should be correct.

Author Response

Dear Reviewer,

We have thoroughly revised the language throughout the manuscript, incorporating the changes highlighted by the reviewer to ensure a smoother and clearer flow. Specifically, we have made significant revisions to the paragraph beginning with "The diverse structures and practices of food hubs..." in order to provide greater clarity on how utilizing a survey methodology adds value to the existing literature on the topic.

We have also addressed the spacing problems identified in the text, ensuring that the updated draft is free of such issues.

We sincerely appreciate the comments provided in the PDF file, and we have addressed each of them accordingly. Regarding the query in the last paragraph of the introduction regarding areas in which food hubs have made positive progress or could improve, we have clarified this in the methodology section. In subsection 2.2, we explain the process of selecting broad indicators of social and environmental sustainability based on previous literature on food system sustainability. Additionally, we demonstrate the linkage between these indicators and the United Nations' Sustainable Development Goals (SDGs) in Table 1.

Regarding the question regarding the criteria for determining a final population size of 150 active food hubs in the United States, we have clarified that we excluded hubs that were no longer in operation or those that did not meet the USDA's definition of food hubs, as presented in the second paragraph of the introduction. Specifically, we excluded hubs that solely offered food processing services, such as kitchens or juice-making facilities.

We genuinely appreciate your thorough review and valuable feedback, which have greatly contributed to improving the clarity and coherence of our manuscript.

Sincerely,

Haniyeh Shariatmadary

on behalf of the team of authors.  

Reviewer 4 Report

This paper is interesting and has intuitive and concrete conclusions.

The title of the article clearly defines the content of the article.

In the abstract, the authors describe the purpose of the study, including relevant information, data collection and evaluation, and clearly state the scope of the experiment/investigation.

The author provides sufficient background information in the introduction to clarify the method and content of this study.

The description of the survey methodology is clear, and the results, discussion, and conclusions are clearly presented and support the point of view.

Please explain the following comments, adjust or supplement the relevant content of your manuscript:

Has the author reviewed the literature on "food safety culture"? If so, please explain the similarities and differences between food safety culture and this study?

How to confirm the “reliability” of the findings of this study? Have you conducted “reliability” and “validity” analysis?

The conclusions are too many and too complicated, whether it can focus on the food safety issues that often occur in Food Hub, such as cross-contamination, temperature control and human negligence etc.

Author Response

Dear Reviewer,

We appreciate your valuable comments and suggestions and your supportive feedback for our paper. Based on your feedback, we have made several important revisions to the manuscript.

First, we have added a new paragraph in Section 3 to discuss the literature on food waste and loss. We acknowledge that there may be some overlaps between the two literatures, particularly in developing economies where food safety and food waste and loss are closely intertwined. However, in advanced countries like the United States, the relationship between these factors is comparatively weaker. While we briefly touch upon these commonalities and differences, we believe that a more dedicated future work exploring these aspects in greater depth would be beneficial.

In addition, we have incorporated hypothesis testing to evaluate each sustainability indicator, taking advantage of our sample size of 30. The small standard errors observed for several indicators have resulted in high Z-scores and narrow confidence intervals. These findings not only reinforce the reliability of our statistical analysis but also provide further support for our research conclusions. We have included this information in a new table within the main text (Table 3), while the previous table has been moved to the appendix (now Table A2).

To enhance the discussion of sample representativeness, we have provided more details in Section 2.1 regarding the sample selection process, specifically highlighting the response rate. Furthermore, in Section 3.1, we emphasize the randomness of our sampling approach, supported by the diversity of food hubs included in our sample and the wide coverage of states represented. Notably, the food hubs in our sample account for 75% of all states with active food hubs, thereby ensuring a robust representation of the broader population.

Lastly, we have condensed and simplified the conclusion section to a single paragraph. This summary provides a high-level overview of our key findings and offers directions for future research.

We sincerely appreciate your valuable input, which has greatly contributed to improving the quality and clarity of our manuscript.

Sincerely,

Haniyeh Shariatmadary

on behalf of the team of authors.  

Reviewer 5 Report

the manuscript describes food hubs (also providing the USDA definition).

the study is based on a survey through which the sustainability of food hubs was evaluated. However, it seems to be based on a self-assessment (as also indicated by the geographical distribution) while some objective indicators measured in the field are missing.

How did the authors verify the reliability of the responses?

Author Response

Dear Reviewer,

We appreciate your valuable comments and suggestions. In response to your concerns, we have incorporated additional clarifications to address them. Our survey study was intentionally designed to target food hub managers who possess the most reliable and comprehensive knowledge about their respective food hubs, including their overall objectives, priorities, and relevant data on sustainability indicators. We made diligent efforts to reach out to all food hubs listed in the USDA directory through email and phone communication. The reported results in our study are based on the responses received within the designated four-month survey window.

Moreover, it is crucial to highlight the geographic coverage of the food hubs included in our sample, which represents a significant proportion of the states where food hubs are present. Specifically, our sample encompasses 75% of all states with active food hubs, ensuring a diverse representation across different regions. This broad coverage enhances the robustness and generalizability of our findings, allowing us to capture a substantial portion of the overall food hub population.

By providing these additional explanations, we aim to reinforce the validity and comprehensiveness of our survey study. The insights and perspectives gathered from food hub managers, who possess in-depth knowledge about their operations and sustainability practices, ensure that our findings accurately reflect the realities of the field.

Thank you once again for your valuable feedback, which has helped us strengthen the clarity and transparency of our study.

Sincerely,

Haniyeh Shariatmadary

on behalf of the team of authors.  

Round 2

Reviewer 3 Report

Fine

Accept in the current form.

Author Response

Thank you. We are glad you found our revisions to the manuscript sufficient.